# Assessing the role of collectivism and individualism on COVID-19 beliefs and behaviors in the Southeastern United States

**Jayur Madhusudan Mehta** [1]*, **Choeeta Chakrabarti**[1], **Jessica De Leon**[2], **Patricia Homan**[3], **Tara Skipton** [4], **Rachel Sparkman**[3]

**1** Department of Anthropology, Florida State University, Tallahassee, Florida, United States of America,
**2** Department of Family Medicine and Rural Health, Florida State University College of Medicine, Tallahassee, Florida, United States of America, **3** Public Health Program, Department of Sociology, Florida State University, Tallahassee, Florida, United States of America, **4** Department of Anthropology, University of Texas, Austin, Austin, Texas, United States of America

* jmehta@fsu.edu

**Data Availability Statement:** All data files are available at https://osf.io/vp3ke/.

## Abstract

America's unique response to the global COVID-19 pandemic has been both criticized and applauded across political and social spectrums. Compared to other developed nations, U.S. incidence and mortality rates were exceptionally high, due in part to inconsistent policies across local, state, and federal agencies regarding preventive behaviors like mask wearing and social distancing. Furthermore, vaccine hesitancy and conspiracy theories around COVID-19 and vaccine safety have proliferated widely, making herd immunity that much more challenging. What factors of the U.S. culture have contributed to the significant impact of the pandemic? Why have we not responded better to the challenges of COVID-19? Or would many people in the U.S. claim that we have responded perfectly well? To explore these questions, we conducted a qualitative and quantitative study of Florida State University faculty, staff, and students. This study measured their perceptions of the pandemic, their behaviors tied to safety and community, and how these practices were tied to beliefs of individualism and collectivism. We found that collectivist orientations were associated with a greater likelihood of wearing masks consistently, severe interruptions of one's social life caused by the pandemic, greater concern for infecting others, and higher levels of trust in medical professionals for behavioral guidelines surrounding the pandemic. These associations largely persist even after adjusting for political affiliation, which we find is also a strong predictor of COVID-19 beliefs and behaviors.

## Introduction

In May 2020, political scientists Carter and May [1] claimed that "limited, politicized notions of COVID-19 as a foreign problem let pass a crucial opportunity to foster a *shared* sense of crisis and need for immediate action across subnational levels" (268; emphasis added). We take this to mean that our opportunity to rise together as a Nation in the face of a pandemic did not

**Funding:** Funding from the Florida State University Libraries Open Access Publishing Fund were made available for publishing this article.

**Competing interests:** The authors have declared that no competing interests exist.

happen–we were not an *exceptional people* in the face of a novel disease and subsequent global pandemic (As of November 2021, Canada experienced roughly 30,000 deaths compared to America's 756,000. Early pandemic news stories described Canadian authorities arresting Americans trespassing and not following safe COVID-19 protocols. What accounts for these drastically different numbers and the dramatically different responses to the pandemic?). Appadurai [2] suggests anthropologists must "observe, nurture, and mobilize" for a new cultural and political moment in which Western nation-states must recognize they are not at the leading edge of pandemic/public-health preparedness. Most compelling is Appadurai's comment that society-at-large is a crucial actor in the mitigation of novel diseases; thus, the "social" is a key component harnessed by the State to mitigate the morbidity and mortality rates of COVID-19. Having lived through this *syndemic* [3], a term that encompasses the social nature of pandemics, we are all acutely aware of the challenges the U.S. faced and that we all encountered in negotiating mask-wearing, social distancing, isolation, and newly developed vaccines.

In this article, we present findings from our mixed methods approach to studying the U.S. response to the global COVID-19 pandemic. We surveyed 3,000 faculty, staff, and students from Florida State University (FSU) at the height of the COVID-19 Delta variant surge in Florida to investigate concepts and trends relative to COVID-19 and collectivism and individualism. This small study is not a representative sample of the US population, yet it provides us with evocative data linking collectivism and practices enhancing group and community safety. In particular, our sample consists of individuals linked to a university, whose mission is public education and research directed towards the public good. Furthermore, this university is in Leon County, a historically democratic population center, and it is likely that our sample universe embodied different values from conservative-leaning population centers in other parts of the state and country [4]. This small sample is a limiting factor certainly, and yet this study has been critical in building new and unique questions about the United States population, health practices, and ideologies of collectivism and individualism.

## Problem orientation / Background

### Previous biosocial and structural research on COVID-19 and the U.S. social and political response

Myth-making [5] in the U.S. about the country's exceptionalism and individuality structured early political narratives of the COVID-19 pandemic, whilst an administration unfriendly to science and academia consistently undermined those doing the work of flattening the curve and keeping U.S. residents alive [1]. In the U.S., we often see ourselves as apart from the rest of the world, but we were not spared the disease, its affects, and its deaths, and we have not been spared its long-term social ramifications [6]. As we experienced the end of the Delta surge and then the subsequent Omicron surge, the United States and its population began facing another set of complications, including rising inflation, scarcity of goods, and a workforce unwilling to return to poverty wages–this again highlights how we are not exceptional in the global sphere [2].

In light of these systemic impacts across all sectors of life, numerous scholars have advocated a holistic study of the COVID-19 pandemic that incorporates structural and social determinants of health and the biosocial linkages across culture, society, and virology. Framed as critical medical anthropology and incorporating the term *syndemic*, medical anthropologists see the utility of incorporating social science, humanities, and the arts into the study of how disease vectors are influenced by politics, economics, and culture [7]. *Syndemic* describes a coupled social and natural disease system, a biosocial explanation of disease that encompasses social and environmental vectors–namely how biology and society intertwine.

Inequality and health inequities exacerbated vulnerability during this pandemic, and made clear how a natural disaster such as this one is anything but natural [8–10]. Failings within the privatized, for-profit health system in the U.S. have served to reinforce structural violence against impoverished, undocumented, and minoritized communities who felt this pandemic differently from wealthier and well-established U.S. residents working jobs we typically call *white collar* [11, 12]. The higher prevalence of comorbidities in impoverished and minority communities, and social determinants of health such as unequal access to healthcare services and resources, and the increased environmental health hazards faced by some communities, all illustrate how health and vulnerability to disease are tied to specific forces of structural inequality [13]. Furthermore, "essential workers" during this pandemic felt their labor elevated to a status of honor, despite facing a multitude of risks, as well as continued harassment and violence stemming from opposition to mask, vaccine, and other COVID-19 mandates designed to reduce transmission and mortality [14–16]. These "essential workers" often faced greater mortality and in which their own health and lives were imperiled to keep society functioning [17, 18].

These structural inequalities have led to the disproportionate impact of COVID-19 on racial and ethnic communities across the globe, including in the U.S. [19, 20]. Socioeconomic factors like economic depression, access to healthy food, limited exposure to pollutants, and housing, and developmental factors like metabolic conditions (i.e., hypertension, diabetes, and obesity), behavioral choices (e.g., smoking, drinking alcohol, etc.), and high cortisol levels have all been demonstrated to be associated with social and structural differences between racial and ethnic communities, leading to health disparities and comorbidities that can increase pandemic risk [19, 21]. Unfortunately, disparate and fragmented public health policies [22] ultimately led to significant mortality rates, as compared to Canada [23] and other nations [24].

## Linking collectivism and Individualism to U.S. pandemic responses

Collectivism describes a condition of society in which the needs of the group are prioritized over the individual; conversely, individualism prioritizes rights, concerns, needs, and desires of each individual [25–29]. Variations in individualism and collectivism are documented across the globe and these variations have influenced nation-state approaches to pandemic preparedness and response. Safe COVID-19 behaviors, including mask wearing and social distancing, were practiced variously and differently across the world. Post-hoc analyses have demonstrated their effectiveness at reducing transmission and mortality [30]. In rural China, increased mask wearing was influenced by a history of rice farming, which required more cooperation and collective coordination than other forms of agriculture [31]. In other cases, collective responses to COVID-19 were structured by national policies and/or unique individual responses [31–33]. In the U.S., where the response to the pandemic was heavily politicized, mask wearing and individual responses to health safety policies were impacted by misinformation, conspiracies, and an active disdain for science [34–38], despite known links between political party affiliation and COVID-19 mortality [39, 40].

Policies emphasizing collective action and the collective good have not been the focus of U.S. development. From inaction on climate change, to crumbling national infrastructure, to disjointed COVID-19 policies, U.S. policies have favored individualism and self-fulfillment over the creation of institutions for the collective good [41, 42]. Countries that were individualistic during the COVID-19 pandemic experienced greater transmission of the disease and mortality from it [43]. Studies have shown that individualist tendencies in a population result in the lack of adherence to safe COVID-19 behaviors and in greater COVID-19 transmission rates [43–46]. These studies demonstrate how individualism negatively impacts health outcomes and

also demonstrate the negative impacts on politically-driven health policy and the impacts of misinformation.

## Methods

This study looks to the collective orientation scale (COS), and evaluates it against demographic data and behavioral responses elicited from our study population to understand the relationships between individualism, collectivism, and social responses to the COVID-19 pandemic in North Florida.

The overarching question that guided development of our study was:

### How do ideologies of individualism and collectivism affect responses to the COVID-19 pandemic?

As our survey responses came in, we developed sub-inquiries into our study:

1. *What demographic parameters predict for collectivism*? With this inquiry, we recognized that numerous forces influence collectivist traits, and we wanted to know if any of our demographic parameters were associated with collectivism, and if so, did our findings match previous studies [47, 48].

2. Does collectivism predict COVID-19 safety behaviors? Numerous studies have shown links between collectivism and safe COVID-19 behaviors [33, 44, 45, 49, 50], and we wanted to test this relationship through our sample and data.

3. Is political affiliation related to COVID-19 safety behaviors and/or collectivism? The political consequences of the pandemic have been described in detail [22, 51, 52], but we wanted to know if political affiliation influenced our study sample in their responses to the pandemic.

### Research design

Our research employed a mixed methods approach to studying the U.S. response to the COVID-19 pandemic and consisted of two phases: 1) qualitative semi-structured interviews, and 2) a quantitative questionnaire. The semi-structured interviews were used to first gain a broad understanding of the beliefs around COVID-19, and to elicit relevant questions and responses for inclusion in the questionnaire; using descriptive interviews to inform the development of a survey tool results in a more grounded approach to data collection [53]. We adapted an approach grounded in cognitive anthropology and psychology [54–56], which was designed to identify culturally salient questions through semi-structured individual interviews and that were re-delivered to a survey population through a questionnaire. Although our methods were not explicitly structured towards identifying cultural domains or cultural consensus, these approaches informed our study design [57–59].

As much of this survey was designed and conducted during the COVID-19 pandemic, planning and implementation of the project was done through the Zoom videoconferencing platform. Phase 1, semi-structured interviews, were conducted in Spring 2021. Interviews (n = 11 participants; 8 female, 3 male) were conducted with an interview guide (S1 File), although the conversations were also allowed to follow salient discussions geared towards helping researchers create potential questions for a quantitative survey for phase 2 of the study. Respondents were gathered using a purposive sampling method. Consent was obtained orally and recorded via Zoom. Semi-structured interviews helped develop portions of the quantitative survey, however, many of our respondents were mostly liberal, and consequently, we needed to be cautious

of how their responses might bias quantitative data collection. The purpose of Phase 1 and this mixed-methods approach was to use respondent answers and ideas to build and design a more purposeful and insightful survey questionnaire. Phase 1 data were not utilized in the analysis of phase 2 data, nor were respondent answers used to assist in the interpretation of phase 2 data. Rather, the semi-structured interviews were simply used as a tool to refine and improve the survey questionnaire, which is the basis for our analysis.

Phase 2 was a quantitative questionnaire that was delivered to respondents in Fall 2021. The quantitative electronic questionnaire (S2 File) was delivered through Qualtrics to a random sample of FSU faculty, staff, and students. The questionnaire consisted of 55 total questions: 13 demographic, 8 qualifying, 7 behavioral, 10 belief-oriented, and 16 for the Culture Orientation Scale (COS) [56] (S3 File). The project team collaboratively developed the questionnaire to answer the project hypothesis and sub-inquiries. Respondents were allowed three total weeks in September 2021 to complete the survey. Cross-group differences in collectivism were assessed using 2-tailed t-tests, and predictors of COVID-19 beliefs and behaviors were assessed using logistic regressions. All analyses were performed using Stata 16. Consent was obtained via an information sheet delivered via Qualtrics which respondents agreed to before proceeding with the study.

Outcome variables for this study include the COS and variables related to COVID-19 beliefs and behaviors. Our collectivism scale (S1 File) was created using the sum of the 16 COS scale items (alpha = 0.67). The theoretical range is 16–144, and the observed range was 61–130. Outcome variables concerning COVID-19 beliefs and behaviors have been dichotomized (yes/no) for logistic regression analyses and include if the respondent wears a mask when leaving their home, if they stay home as much as possible to mitigate spread of COVID-19, if they are vaccinated, if their socializing has been severely impacted by the pandemic, if their daily routine has been severely impacted by the pandemic, and if they strongly agree with the statement: "I am worried about the possibility of infecting others." Similarly, outcome variables concerning trust in COVID-19 information have been dichotomized (yes/no) for logistic regression analyses and include if the respondent trusts that they can get accurate information regarding COVID-19 and the pandemic and if they trust medical professionals regarding COVID-19 safety guidance and vaccine safety.

Covariates for this study include the respondent's gender (female/male), their age (measured in years), race (White/not White), if they are a college graduate (yes/no), and their political affiliation (Republican, Democrat, Independent, or Other). Household income was assigned categorically into one of the following four income groups: less than $30,000; $30–69,000; $70–99,000; and $100,000 and higher.

### Ethics statement

The study was approved by the Florida State University Institutional Review Board (IRB; Study 02064)) and conducted in accordance within the guidelines of the American Anthropological Association Principles of Professional Responsibility [60]. All participants were informed of the research and were provided an information sheet to review before choosing to participate in the study voluntarily.

### Results

In our analyses, we made the following interpretations of our data– 1) We found that collectivism is strongly tied to education and older age. 2) We also found that political affiliation was a good predictor for safe COVID-19 practices/behaviors. Furthermore, we found that 3) Trust in medical professionals was also tied to collectivism. Consequently, 4) We found that the

Collectivism scale was effective at predicting practices we would associate with defeating the COVID-19 pandemic [33]–and our very liberal, college-educated sample was highly compliant with these practices. However, for those that are more individualistic as well as politically conservative and Republican, we found that they had less trust in science/medicine, and they participated less in mask wearing, social distancing, and vaccination. As Grossman et al. [61] have identified, partisanship among political leaders [62] strongly predicted COVID-19 prevention behaviors and, thus, voluntary choices by individuals. What our study shows is that some individuals who self-identify as republican tended to not trust medical professionals or to participate in preventive COVID-19 behaviors, nor get vaccinated. We must however note that our sample of self-identified republicans (12%) is quite small, and future studies could strive to oversample for underrepresented categories in the study population.

Of the 3,000 FSU affiliates polled, we received n = 283 total responses in return, yielding a response rate of 9.4%. After removing incomplete responses, 251 complete cases were used for regression analysis. While a higher response rate would be desirable, our response rates are consistent with (and in many cases higher than) what is typical for online studies which have recently published results in peer-reviewed scientific outlets [63–65]. It is likely that the mental exhaustion of most individuals during the COVID-19 pandemic, and the overwhelming amount of time individuals spent on their computers, may have lowered the response rate of this survey, but it is not clear how this would affect the results. Table 1 shows the composition of our sample compared to the U.S. population, based on 2019 American Community Survey data and PEW Research Center [66] data on political affiliation. Overall, our sample has more females (67%), is highly educated (73%), and identify the most as Democrats (58%), White (86%), and have higher incomes than the overall U.S. population.

Table 2 shows the descriptive statistics for study variables that include respondents' behaviors and viewpoints concerning COVID-19 and health safety. In our sample, 93% are vaccinated compared to the national rate of 59% [67] and the Florida rate of 60.6% [68]. It is likely the ready availability of vaccines administered by FSU influenced the high rate of vaccination of our population. Regarding their individual behaviors, 73% of the sample reports wearing a mask when leaving their home, while 48% tries to stay home as much as possible. Slightly more than half (52%) of the sample reports that socializing has been severely impacted by the

**Table 1. Sample composition compared to US population.**

|  | Study Sample | U.S. Population |
|---|---|---|
| Female | 67% | 51% |
| Median Age | 40 | 38 |
| College Graduate | 73% | 32% |
| White | 85% | 76% |
| Median HH Income | $85k | $63k |
| Democrat | 58% | 33% |
| Republican | 12% | 29% |
| Independent | 16% | 34% |
| Other/unaffiliated | 14% | 4% |
| *N* | 251 |  |

Table 1 compares our sample to the U.S. population. Our sample was more female, more highly educated, more liberal, more ethnically White, and had higher incomes that the overall US population. Population estimates are based on 2019 ACS data, except political affiliation, which is derived from PEW Research center [66] (https://pewrsr.ch/2TpQBnx).

**Table 2. COVID behaviors and beliefs of sample (N = 251).**

| STUDY VARIABLE | PERCENTAGE |
|---|---|
| High Risk Category | 34% |
| Vaccinated | 93% |
| Always Mask When Leaving Home | 73% |
| Stay Home as Much as Possible | 48% |
| Concerned About Infecting Others | 48% |
| Daily Routine Severely Impacted | 42% |
| Socializing Severely Impacted | 52% |
| Trust I Can Get Accurate Information | 37% |
| Trust Medical Professionals for COVID Safety Guidance | 54% |
| Trust Medical Professionals for COVID Vaccine Safety | 59% |
| Strongly Agree COVID-19 Vaccines are safe | 70% |

Table 2 shows the percentage of respondents who engaged in specific COVID safety behaviors or espoused specific COVID-related beliefs. The vast majority of the sample is vaccinated, believes COVID vaccines are safe and always work a mask when leaving home at the time of the survey. The other beliefs and behaviors are more split, with roughly half of respondents adhering.

pandemic, and less than half (42%) have had their daily routine severely impacted. While a notable portion of the sample does not seem to trust in media (37%), most have trust in medical professionals (54% and 59%), and 70% strongly agree that COVID-19 vaccines are safe. On the collectivism scale, the mean score was 91.4 (S.D. = 11.8). This means that per question, respondents averaged 5.7, which as a scalar value, lands between neither agree or disagree (5) and mildly agree (6) on the COS scale items. Consequently, our sample average is just slightly more to the collectivist side of center/neutral.

In terms of our demographic variables, several are associated with collectivism. First, age is a strong predictor for collectivism (b = 0.240, $p < 0.001$), with every 10-year increase of age being associated with an increase of 2.4 units on the collectivism scale. Fig 1 shows mean collectivism scores (and 95% confidence intervals) for each of the demographic groups in our sample. Women and college graduates are more collectivist than men and non-graduates ($p < 0.05$), and non-white respondents are more collectivist than White respondents, although this difference was only marginally significant (p = 0.089) due to the small sample size of racial/ethnic minorities. Hui and Yee report a positive association between age and collectivism, as well as between sex (females) and collectivism [47]. Vandello and Cohen report Florida in the 2nd quintile for collectivism due to the age of the population and a larger than average immigrant population [48]. Collectivism scores do not differ significantly according to political affiliation.

We created a series of nested models (Table 3) to determine the impact of collectivism and demographic covariates on COVID-19 behaviors. Model 1 includes age and if the respondent is high risk because these are crucial predictors of both health outcomes and health behaviors in general and particularly in the case of COVID-19. Model 2 adds being Republican as a predictor given the politicized nature of COVID-19, and Model 3 includes if the respondent is female to determine if gender has an impact on collectivism and COVID-19 behaviors. Race was not included as a covariate due to the small sample size of non-White respondents (n = 35), but supplemental analyses showed the results did not differ meaningfully when it was included. Due to the high vaccination rate of our sample (93%), we conducted supplemental analyses (see S4 File) using the Firth Method for logistic modeling to account for the small number of unvaccinated respondents. Results using the Firth Method do not substantively

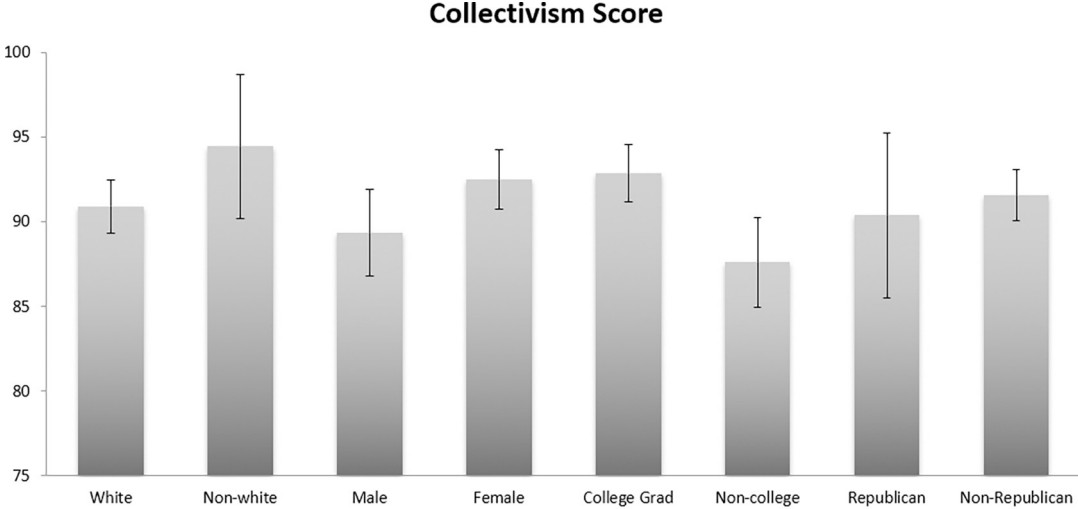

**Fig 1. Mean collectivism scores by demographic groups with 84% confidence intervals.** Fig 1 shows the mean Collectivism scores for various demographic groups in the study, with 84% confidence intervals which are appropriate for visualizing differences that are significant at the p<0.05 level [69]. The theoretical range of the Collectivism scale is 16–144, and the observed range was 61–130. Female respondents are more collectivist than male respondents and those with a college degree are more Collectivist that those without a degree, and these differences are significant at the p < .05 level (two-tailed tests). Race difference is marginally significant at p < .10, which does suggest a meaningful difference given the small sample size of non-white respondents.

**Table 3. Logistic regression on COVID behaviors by collectivism (N = 251).**

| Outcome | Mask Wearing | | | Vaccinated | | | Socializing Severely Impacted | | | Concerned about Infecting Others | | |
|---|---|---|---|---|---|---|---|---|---|---|---|---|
| Covariate | Model 1 | Model 2 | Model 3 | Model 1 | Model 2 | Model 3 | Model 1 | Model 2 | Model 3 | Model 1 | Model 2 | Model 3 |
| Collectivism | 0.034* | 0.036* | 0.030+ | 0.010 | -0.003 | -0.003 | 0.024* | 0.023+ | 0.021+ | 0.025* | 0.024* | 0.023+ |
| | (0.013) | (0.015) | (0.016) | (0.022) | (0.025) | (0.026) | (0.012) | (0.012) | (0.012) | (0.012) | (0.012) | (0.012) |
| Age | -0.008 | -0.012 | -0.009 | 0.030 | 0.031 | 0.031 | 0.016 | 0.016 | 0.017 | -0.032** | -0.035** | -0.035** |
| | (0.014) | (0.015) | (0.016) | (0.024) | (0.029) | (0.030) | (0.012) | (0.012) | (0.012) | (0.012) | (0.013) | (0.013) |
| High Risk | -0.362 | 0.083 | 0.200 | 0.348 | 1.253 | 1.252 | -0.664+ | -0.496 | -0.476 | 0.433 | 0.689+ | 0.709+ |
| | (0.410) | (0.475) | (0.493) | (0.793) | (1.015) | (.016) | (0.374) | (0.381) | (0.381) | (0.373) | (0.392) | (0.395) |
| Republican | | -3.307*** | -3.338*** | | -3.226*** | -3.227*** | | -1.536** | -1.506** | | -2.343*** | -2.320*** |
| | | (0.584) | (0.599) | | (0.627) | (0.635) | | (0.488) | (0.490) | | (0.634) | (0.635) |
| Female | | | 1.147** | | | -0.009 | | | 0.225 | | | 0.181 |
| | | | (0.344) | | | (0.620) | | | (0.286) | | | (0.293) |
| Constant | -1.687 | -1.363 | -1.651 | 0.452 | 2.229 | 2.228 | -2.552 | -2.316 | -2.375* | -1.250 | -0.906 | -0.954 |
| | (1.178) | (1.321) | (1.368) | (1.942) | (2.211) | (2.211) | (1.056) | (1.083) | (1.088) | (1.038) | (1.086) | (1.091) |

Standard errors are in parentheses. _Constant represents the Y intercept.

+p < .10

*p < .05

**p < .01

***p < .001.

Table 3 shows the results of a series of logistic regressions predicting four types of COVID-related behaviors as a function of Collectivism scores and demographic characteristics. Collectivism is positively associated with all behaviors except vaccination. Republican political affiliation is negatively associated with all studied COVID behaviors.

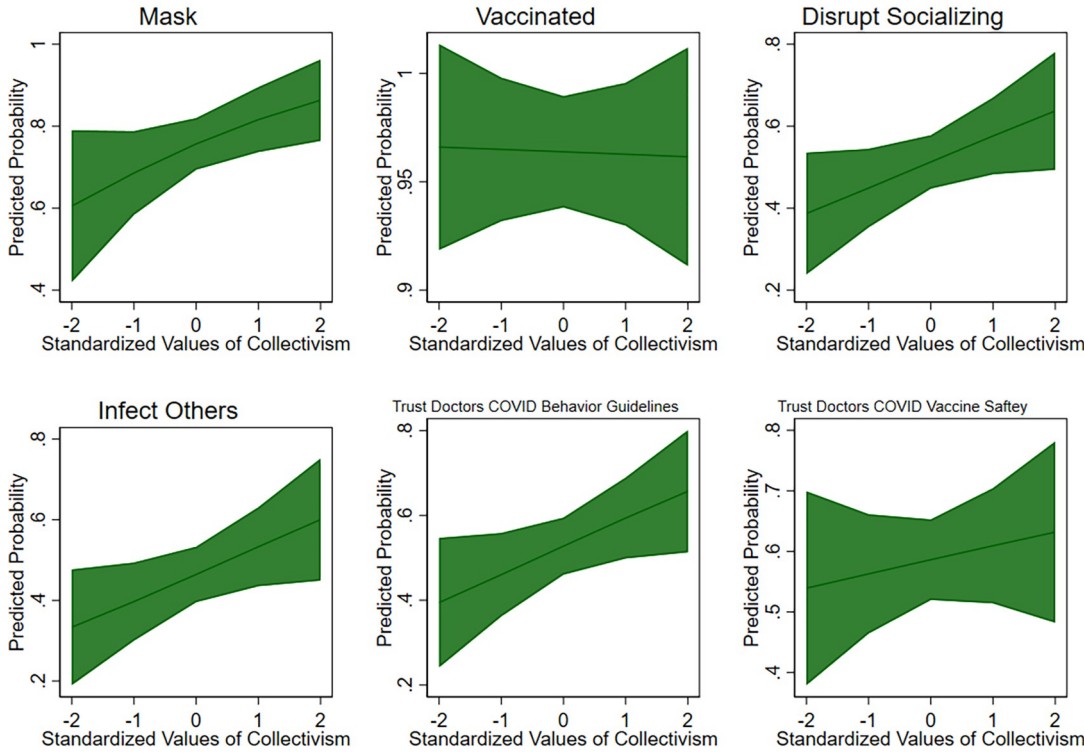

**Fig 2. Collectivism and COVID-19 safety behaviors.** Fig 2 shows the predicted probabilities (calculated based on Model 3 in Tables 3 and 4) of a variety of COVID-19 behaviors and beliefs as a function of collectivism, while holding age, gender, high risk status and political affiliation constant at their means. The figures demonstrate that higher collectivism is associated with higher probabilities of all of the COVID-19 safe behaviors and beliefs except being vaccinated.

differ from those using a standard logistic modelling approach, and we therefore use the standard approach in our main analyses (Table 3, Figs 2 and 3) for simplicity, consistency across outcomes, and ease of interpretation. Concerning trust in medical professionals, Table 4 uses the same sets of covariates to regress collectivism on trust in doctors. We found that collectivism strongly predicts the following outcomes: mask wearing, disrupted socializing, concern for infecting others, and trust in doctors regarding COVID-19 behavior guidelines (see Fig 2, and Tables 3 and 4).

## Discussion

Our inquiry began with the question, *how does an individual's alignment along the collectivism/ individualism scale impact their choices regarding the COVID-19 pandemic?* Through a quantitative survey of FSU affiliates, we discovered age, sex, and education are associated with collectivism, and that individuals that are more collectivist also tend to wear masks, practice social distancing, have concerns for infecting others, and trust in doctors regarding COVID-19 behavior guidelines. Given our questionnaire also asked respondents questions about political affiliation, we wanted to unpack how and why political beliefs influenced our study samples, since numerous studies have shown these to be interlinked phenomena [52].

Compared to those who do not identify as Republican (i.e., Democrats, Independents, and other political affiliations), we found that self-identified Republicans were highly unlikely to do the following (significant at p < .05; Fig 3): mask wearing, disrupt socializing, have concern

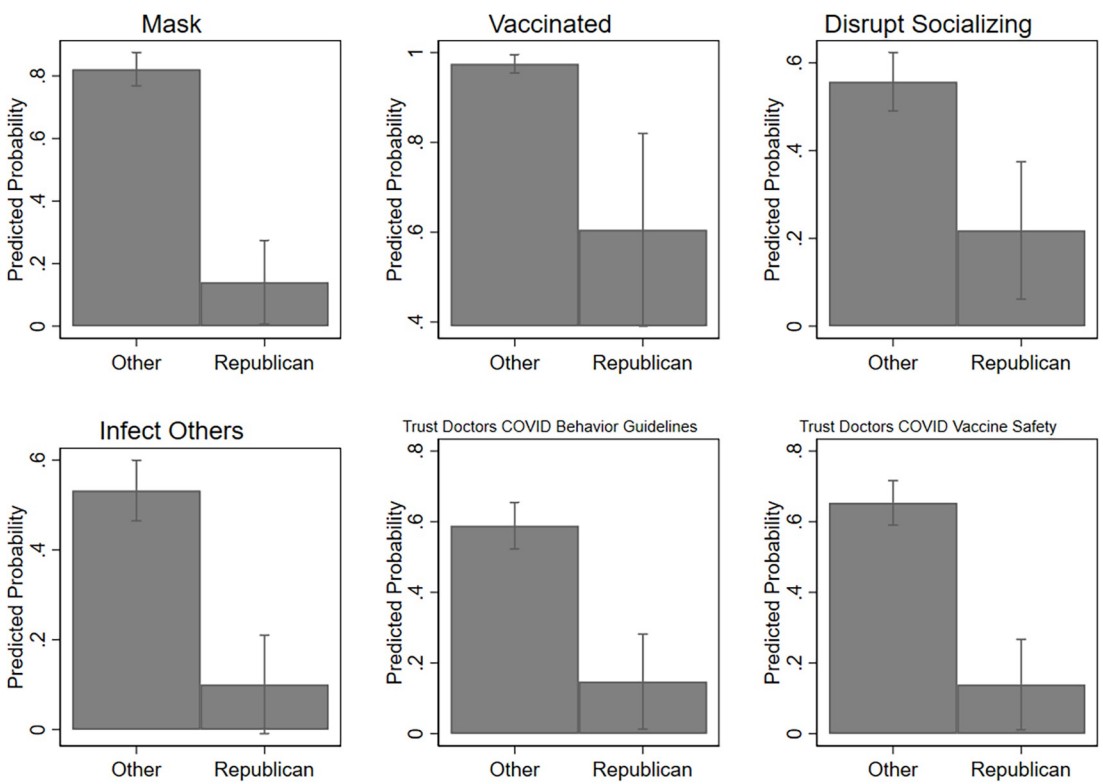

**Fig 3. Political identity and COVID-19 safety behaviors.** Fig 3 shows the predicted probabilities (calculated based on Model 3 in Tables 3 and 4) of a variety of COVID-19 behaviors and beliefs among Republicans versus those with other political affiliations (including Democrat, Independent and other), while holding age, gender, high risk status and collectivism constant at their means. The figures demonstrate that Republican are significantly less likely than other groups to consistently wear as mask, to be vaccinated, to have allowed the pandemic to disrupt their socializing, to be concerned about infecting others, and to trust doctors for behavioral guidance and vaccine safety information.

about infecting others, and trust medical professionals concerning COVID-19 behavior guidelines or vaccine safety. The relationship between political affiliation and public health practice does not seem necessarily clear, yet numerous studies have demonstrated links between political party, political affiliation, and perspectives on the COVID-19 pandemic [51, 70–77]. It is likely that the significant politicization of this novel disease [22, 52] resulted in mistrust in health professional and scientifically-informed public health policy. A future study might try to parse *why* segments of our study population did not trust medical health policy guidance. While collectivism does predict for safe COVID-19 behaviors like mask wearing and trust in health policy guidance, it does not predict for political affiliation.

For those that provided written responses for why they chose to *not* get vaccinated, many did not trust its efficacy or entertained untruths about the vaccine itself, including that it was only a money-maker for pharmaceutical companies, that it was made from aborted fetal cells, and that it caused infertility. While we cannot answer why the respondents felt this way, a future study might look at a targeted sub-sample of a population that has significant vaccine hesitancy to evaluate if this group is particularly individualist or collectivist, or if they are particularly conservative or liberal. In this sample, only 18 respondents provided a reason for not getting vaccinated. Of these 18 respondents, 10 of them report being a Republican, 2 Democrat, 2 Independent, and 4 in the "Other" category.

Finally, in terms of our study population, we should not think of our sample as a heavily skewed set of professors. Many of the faculty, staff, and students that work at FSU come from a

**Table 4. Logistic regression on trust in doctors by collectivism (N = 251).**

| Outcome | Trust Doctors Behavioral Guidelines | | | Trust Doctors Vaccine Safety | | |
|---|---|---|---|---|---|---|
| Covariate | Model 1 | Model 2 | Model 3 | Model 1 | Model 2 | Model 3 |
| Collectivism | 0.0258* | 0.024+ | 0.023+ | 0.011 | 0.009 | 0.008 |
| | (0.012) | (0.012) | (0.012) | (0.012) | (0.012) | (0.012) |
| Age | 0.009 | 0.008 | 0.009 | 0.008 | 0.008 | 0.008 |
| | (0.012) | (0.012) | (0.012) | (0.012) | (0.013) | (0.013) |
| High Risk | -0.642+ | -0.434 | -0.414 | -0.386 | -0.117 | -0.109 |
| | (0.372) | (0.383) | (0.384) | (0.371) | (0.391) | (0.392) |
| Republican | | -2.143*** | -2.114*** | | -2.470*** | -2.457*** |
| | | (0.563) | (0.565) | | (0.563) | (0.565) |
| Female | | | 0.231 | | | 0.088 |
| | | | (0.290) | | | (0.298) |
| Constant | -2.286* | -2.013+ | -2.078+ | -0.858 | -0.416 | -0.439 |
| | (1.053) | (1.099) | (1.105) | (1.037) | (1.104) | (1.107) |

Standard errors are in parentheses. Constant represents the Y intercept.

+$p < .10$

*$p < .05$

**$p < .01$

***$p < .001$.

Table 4 shows the results of a series of logistic regressions predicting two types of COVID-related beliefs as a function of Collectivism scores and demographic characteristics. Collectivism is positively associated with trusting doctors for behavioral guidelines. Republican political affiliation is negatively associated with both aspects of trust in doctors.

wide-ranging territory in the surrounding rural counties and from across Florida, which can be variously conservative and/or liberal. Only 58% of our sample respondents self-identified as Democrat. The study sample is not ethnically diverse and skews white, well-educated, and more affluent than the national average. Our respondents were also neither overwhelmingly collectivist nor were they highly individualistic. As indicated above, our sample average is just slightly more to the collectivist side of center/neutral. Future studies should strive to identify a greater diversity of respondents, where they were raised and where they live presently, and how their background influences collectivism and health safety practices. While we cannot use this study and/or this study population to make generalized claims about the US population writ large, we do have the ability to discuss fascinating patterns in the data. In particular, what factors influenced our 93% vaccination rate? Furthermore, whether collectivist or individualist, more than half (73%) reported wearing masks, indicating that consistent and pervasive health policy messaging was positively impacting safe COVID-19 behaviors. Understanding why is still more challenging yet, and additional studies are needed to elucidate these fascinating trends.

## Impacts of the pandemic and the future

We also asked our respondents what they thought about the future. When asked "if the pandemic has brought people together?", 43% of faculty and 39% of staff strongly disagreed. However, only 22% of students disagreed. How did students experience this pandemic differently than older faculty and staff? Is it because younger students live their worlds online? Is it because they did not really follow social distancing and quarantine guidelines? These are questions we intend to continue asking of the study dataset and in future analyses. We also asked if these pandemic-related changes would continue into the future, and all sub-groups agreed

that U.S. society had changed permanently. It would be interesting to ask what changes in particular they perceive to be permanent. We also asked if the U.S. is more divided: faculty and staff both answered at 48% strongly agree, the U.S. is more divided now; 38% of students agree that the U.S. is more divided now than before the pandemic. From this line of questioning, younger students appear more optimistic than faculty and staff, and perhaps we can take some lessons from the younger generation about how to perceive this pandemic and how we as a society have responded to it.

Returning to our original inquiry about the association of individualism on our response to the pandemic, our study demonstrated that collectivist individuals practiced safe COVID-19 protocols, suggesting that they accepted the Center for Disease Control's (CDC) protocols and recommendations to get vaccinated. Those who scored high for individualism on the collective orientation scale (COS) were not associated with any specific political affiliation. Individuals who identified as conservative or Republican were more likely not to follow COVID-19 safety protocols. It is likely that the politicization of the pandemic in its early days played a significant role in how individuals perceived their efficacy of social isolation, mask-wearing, and vaccines. Variable and disjointed policies and lack of enforcement certainly may have contributed to our excessive mortality rates [10], likely also influenced by uniquely American patterns of health [78–81]. The rural and urban divide certainly impacted mask-wearing and mask prevalence in parts of China, where mask-wearing was variously adopted according to government measure, population density, and community practice [31–33]. How might we understand differences in rural and urban health practices and COVID-19 outcomes in the United States? Numerous studies have indicated that rurality impacts safe COVID-19 practice due to increased age, political ideology, and community experiences tied to population density [82–84]. Our study demonstrates practices of a relatively urban population surrounded by a rural community, and future studies in North Florida should seek to disentangle rural/urban practices and their relationships to individualism and collectivism.

While we cannot use our data to generalize to the United States population as a whole, especially since our study sample is mostly a well-vaccinated group of cooperative individuals who were complaint with COVID-19 public health guidelines, we can offer a few summary thoughts here:

1. The COVID-19 pandemic is more accurately described as a *syndemic*, given the social determinants of health, disease and mortality that have greatly impacted its spread.

   a. In the U.S., minority and marginalized groups experienced greater mortality from COVID-19 during this pandemic as compared to affluent U.S. residents.

   b. Our study population was not sufficiently diverse to evaluate variable expressions of COVID-19 behaviors

2. In a national study, Celinska [85] found that minorities and individuals with lower socio-economic status tended to be more collectivistic.

   a. We found that age and education best predicted for collectivism, not race or lower SES, however, our sample was not well represented by minorities or individuals with lower SES.

Future studies should perform cultural domain analysis on topics such as pandemic behaviors, individualism/collectivism, and political affiliation to get a more detailed understanding of how new sources and politics have real world, mortality-centric implications. It remains unclear how much our small, liberal, and well-educated study population has influenced our analysis, and consequently, future studies should employ a large, more diverse sample set, as

well as employ ethnographic interviews. Future research should also use a larger sample to explore the role the media plays in guiding public opinion and how public policy is influenced by political processes. In particular, we are curious to learn how U.S. residents consider themselves along the COS, and what role they think that plays in the U.S. response to the COVID-19 pandemic/*syndemic*.

## Supporting information

**S1 File. Interview guide.** Qualitative Information Guide.
(PDF)

**S2 File. Quantitative survey.** Available online at https://osf.io/vp3ke/
(PDF)

**S3 File. Culture orientation scale.** Evaluation tool for collectivism/individualism.
(PDF)

**S4 File. Firth method data.** Comparison of Results from Traditional Logistic Regression and Those adjusted for Rare Events Using the Firth Method.
(PDF)

## Acknowledgments

We thank our colleagues in the Departments of Sociology, Anthropology, and the College of Medicine at Florida State University. The FSU Office for Human Subjects Protection and the Office of Institutional Research facilitated this research and made data collection seamless. The UT-Austin Anthropology Department also deserves recognition, as do Jayant Mehta (ETSU College of Medicine) and Jim Whyte (FSU College of Nursing), who provided valuable feedback on an early draft of this paper. An early draft of this paper was presented at the 2021 American Anthropological Association Conference and many of the panel participants gave exciting feedback, including Raja Swamy (University of Tennessee) and Greg Thompson. Nora Haenn (North Carolina State University) also encouraged Mehta to continue this line of research and deserves recognition as well. We also wish to thank two anonymous reviewers whose comments substantially assisted in improving the manuscript.

## Author Contributions

**Conceptualization:** Jayur Madhusudan Mehta, Jessica De Leon, Tara Skipton.

**Data curation:** Jayur Madhusudan Mehta.

**Formal analysis:** Choeeta Chakrabarti, Jessica De Leon, Patricia Homan, Tara Skipton, Rachel Sparkman.

**Investigation:** Jayur Madhusudan Mehta, Choeeta Chakrabarti, Jessica De Leon, Patricia Homan, Tara Skipton.

**Methodology:** Jayur Madhusudan Mehta, Choeeta Chakrabarti, Jessica De Leon, Patricia Homan, Tara Skipton, Rachel Sparkman.

**Project administration:** Jayur Madhusudan Mehta.

**Visualization:** Patricia Homan, Rachel Sparkman.

**Writing – original draft:** Jayur Madhusudan Mehta, Choeeta Chakrabarti, Jessica De Leon, Patricia Homan, Tara Skipton.

**Writing – review & editing:** Jayur Madhusudan Mehta, Choeeta Chakrabarti, Jessica De Leon, Patricia Homan, Tara Skipton, Rachel Sparkman.

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
