## [Decision Letter · Decision Letter 0]

14 Jul 2022

PONE-D-22-14363Assessing the Role of Collectivism and Individualism on COVID-19 Beliefs and BehaviorsPLOS ONE

Dear Dr. Mehta,

Thank you for submitting your manuscript to PLOS ONE. After careful consideration, we feel that it has merit but does not fully meet PLOS ONE’s publication criteria as it currently stands. Therefore, we invite you to submit a revised version of the manuscript that addresses the points raised during the review process.

We look forward to receiving your revised manuscript.

Kind regards,

Abbas Al Mutair

Academic Editor

PLOS ONE

Journal Requirements:

Reviewers' comments:

Reviewer's Responses to Questions

**Comments to the Author**

1. Is the manuscript technically sound, and do the data support the conclusions?

Reviewer #1: Partly

Reviewer #2: Yes

2. Has the statistical analysis been performed appropriately and rigorously? 

Reviewer #1: Yes

Reviewer #2: Yes

3. Have the authors made all data underlying the findings in their manuscript fully available?

Reviewer #1: No

Reviewer #2: No

4. Is the manuscript presented in an intelligible fashion and written in standard English?

Reviewer #1: Yes

Reviewer #2: No

5. Review Comments to the Author

Reviewer #1: This article provides an interesting perspective into the challenges and experiences of Americans during the COVID-19 pandemic. The manuscript explores perceptions of behaviours and attitudes that how certain collective or individualistic orientations might foster less or more mask use. This paper has potential but needs major revision to be considered.

1. The structure of the paper is abnormal. Various areas in the literature review at unconventional and do not follow typical social science articles. Please follow typically PLoS ONE articles.

2. This literature is incomplete and does not include reviews of some important collectivism-mask papers that have made important contributions.

Lu et al., 2021 PNAS,

English & Li 2021 Frontiers in Psychology

English et al., 2022 Current Research in Ecological and Social Psychology

3. Methodology has major weaknesses. While it is important to recognize the low response rate as there were more than 3,000 participants surveyed, but only 283 participants completed the survey. Second, I am confused by the role of the interview study. While I value mixed method approaches, I am uncertain what value or contribution it makes in the study.

Please clarify.

4. It remains how this study at FSU can be generalized to the US population. Florida represents a special state that has taken an a unique path in combating the coronavirus. Authors must reframe their conclusions and interpretations with caution given the local context. In fact, authors might was to conduct further analyses to explain where people are from might have an interactive effect on their attitudes.

Authors should post their Data and code on OSF with a permanent link. Not a wordpress site.

Reviewer #2: Review: Assessing the role of collectivism and individualism on COVID-19 Beliefs and Behaviors

The study addresses the important question of what factors of the U.S. culture have contributed to the significant impact of the pandemic by means of a qualitative and quantitative study of Florida State University faculty, staff and students. Specifically, the study measured their perceptions of the pandemic, their behaviors and how these practices were tied to beliefs of individualism and collectivism. The authors found that collectivist orientations were associated with larger adherence to preventive measures, greater concern for infecting others, and higher trust in medical professionals.

The authors claim that their aim would be to study the U.S. response to the global COVID-19 pandemic. I do not think that such a claim can be made with such a small and non-representative sample. In fact, the relatively small and non-representative sample (University population) is a big limitation of the study, this should already be mentioned in the introduction so that the readers can adapt their expectations accordingly.

Also, the proportion of Republican was rather small (12%), which limits conclusions about political associations. These limitations should be discussed more explicitly.

I was also wondering why the second paragraph of the introduction summarises the results, this should be reported in the results section.

Regarding the background information, I found that the style of writing addresses quite a specific readership (antropology?). As a social scientist, I had a hard time reading this paragraph and understanding what this study is about. To make the study more accessible, I suggest that the authors focus more strongly on information that is relevant for the current research question, and devote more time to studies that addressed similar questions.

For example, there are many studies that addressed how individual differences and political attitudes shape the perception of the pandemic and the preventive behaviour, some of them also focusing on individualism vs. collectivism. The introduction does not sufficiently take these previous studies into account and the scientific merit of the current study remains unclar without properly link the present study to previous research. Some examples are presented below.

Analysis:

I am not sure whether it make sense to run a logistic regression with n=251 for vaccinated when 93% of people were vaccinated. If anything, the authors should consider a zero-inflated logistic regression model (with vacc recoded as zero).

It should be explained explicitly why age was included in Model 1 but gender in Model 3 (I think it makes sense but it should be explained).

Table 3: it should be specified in the table notes what _cons means.

Table 4: it should be specified in the table notes what femalefm and _cons means.

Figure 3: Figure captions are missing; it should for example be specified what “Other” means.

There is a strange style of the references.

Some examples of suggested references that should be cited:

Earnshaw, V. A., Eaton, L. A., Kalichman, S. C., Brousseau, N. M., Hill, E. C., & Fox, A. B. (2020). COVID-19 conspiracy beliefs, health behaviors, and policy support. Translational Behavioral Medicine, 10(4), 850–856.

Gratz, K. L., Richmond, J. R., Woods, S. E., Dixon-Gordon, K. L., Scamaldo, K. M., Rose, J. P., & Tull, M. T. (2021). Adherence to social distancing guidelines throughout the COVID-19 pandemic: The roles of pseudoscientific beliefs, trust, political party affiliation, and risk perceptions. Annals of Behavioral Medicine, 55(5), 399–412.

Hartmann, M., & Müller, P. (2022). Acceptance and Adherence to COVID-19 Preventive Measures are Shaped Predominantly by Conspiracy Beliefs, Mistrust in Science and Fear–A Comparison of More than 20 Psychological Variables. Psychological reports, 00332941211073656;

Maaravi, Y., Levy, A., Gur, T., Confino, D., & Segal, S. (2021). “The tragedy of the commons”: How individualism and collectivism affected the spread of the COVID-19 pandemic. Frontiers in public health, 9, 627559.

Rajkumar, R. P. (2021). The relationship between measures of individualism and collectivism and the impact of COVID-19 across nations. Public Health in Practice, 2, 100143.

Šrol, J., Ballová Mikušková, E., & Čavojová, V. (2021). When we are worried, what are we thinking? Anxiety, lack of control, and conspiracy beliefs amidst the COVID‐19 pandemic. Applied Cognitive Psychology, 35(3), 720–729)

Xiao, W. S. (2021). The Role of collectivism–individualism in Attitudes toward compliance and Psychological responses during the COVID-19 pandemic. Frontiers in Psychology, 12, 600826.

Zajenkowski, M., Jonason, P. K., Leniarska, M., & Kozakiewicz, Z. (2020). Who complies with the restrictions to reduce the spread of COVID-19?: Personality and perceptions of the COVID-19 situation. Personality and individual differences, 166, 110199.

6. PLOS authors have the option to publish the peer review history of their article (what does this mean?). If published, this will include your full peer review and any attached files.

Reviewer #1: No

Reviewer #2: No

---

## [Author Response · Author response to Decision Letter 0]

30 Aug 2022

Thank you for reading our manuscript closely and providing feedback and guidance. Our work has improved immensely due to your suggestions.

---

## [Decision Letter · Decision Letter 1]

16 Nov 2022

PONE-D-22-14363R1Assessing the Role of Collectivism and Individualism on COVID-19 Beliefs and BehaviorsPLOS ONE

Dear Dr. Mehta,

Thank you for submitting your manuscript to PLOS ONE. After careful consideration, we feel that it has merit but does not fully meet PLOS ONE’s publication criteria as it currently stands. Therefore, we invite you to submit a revised version of the manuscript that addresses the points raised during the review process.

We look forward to receiving your revised manuscript.

Kind regards,

Angelo Moretti, Ph.D.

Academic Editor

PLOS ONE

Journal Requirements:

Reviewers' comments:

Reviewer's Responses to Questions

**Comments to the Author**

1. If the authors have adequately addressed your comments raised in a previous round of review and you feel that this manuscript is now acceptable for publication, you may indicate that here to bypass the “Comments to the Author” section, enter your conflict of interest statement in the “Confidential to Editor” section, and submit your "Accept" recommendation.

Reviewer #1: All comments have been addressed

Reviewer #2: (No Response)

2. Is the manuscript technically sound, and do the data support the conclusions?

Reviewer #1: Yes

Reviewer #2: Yes

3. Has the statistical analysis been performed appropriately and rigorously? 

Reviewer #1: Yes

Reviewer #2: Yes

4. Have the authors made all data underlying the findings in their manuscript fully available?

Reviewer #1: Yes

Reviewer #2: Yes

5. Is the manuscript presented in an intelligible fashion and written in standard English?

Reviewer #1: Yes

Reviewer #2: Yes

6. Review Comments to the Author

Reviewer #1: I support the publication of this manuscript. It has improved substantially. I would ask the authors to revise their title to into "in the Southeastern United States" because I believe a lot of key findings are very context specific to Florida. This will also be valuable for future readers who will ponder the unique COVID response in Florida as early as May 2020. It is important to note that college represents a diverse group of students and teachers who might share different opinions and attitudes from local authorities who removed mask policies at that time. It might be worth expanding on this as well.

Reviewer #2: I think the article improved greatly in the revised version.

There are still some remaining issues which should be improved:

- There is a section in the introduction with the title “Previous research on COVID-19 and the U.S. Social and Political Response”. What is listed in this four short review paragraphs is only a very small snapshot of what has been researched on this topic, and it was not clear to my how the author selected the content. I think the title of the paragraph does not match the content. The authors should state explicitly at the beginning of this section that they selected some results that they think are relevant for the present research, or the title of the paragraph should me more specific.

- A formal definition of collectivism and individualism should be provided

- At the end of the introduction, the authors list some more research questions (is political affiliation related to COVID-19 safety behaviors; relationship between collectivism and trust in science). The authors should at least briefly introduce these questions with the idea behind it and the relevant references that addressed these questions already.

- The first paragraph of the results section summarizes/discusses all relevant interpretations of the results. This is unusual, the results should be reported first.

- A substantial part of the discussion contains results. The structure should be improved. The discussion should start with discussing the results of the main question, not with additional results

- There should be a participant section in the method section, where one can read number and demographic information (age, gender) of the participants for the different surveys. Were there really only 11 participants who conducted the semi-structured interview? Such limitations should also be mentioned in the discussion section.

7. PLOS authors have the option to publish the peer review history of their article (what does this mean?). If published, this will include your full peer review and any attached files.

Reviewer #1: No

Reviewer #2: No

---

## [Author Response · Author response to Decision Letter 1]

22 Nov 2022

Thank you for reading our manuscript and providing detailed and useful comments. 

OVERALL REVISION COMMENTS – ROUND 2 – 11/16/2022

PONE-D-22-14363R1

Assessing the Role of Collectivism and Individualism on COVID-19 Beliefs and Behaviors

RESPONSES TO THE REVIEWERS

Comments to the Author

1. If the authors have adequately addressed your comments raised in a previous round of review and you feel that this manuscript is now acceptable for publication, you may indicate that here to bypass the “Comments to the Author” section, enter your conflict of interest statement in the “Confidential to Editor” section, and submit your "Accept" recommendation.

Reviewer #1: All comments have been addressed

Reviewer #2: (No Response)

2. Is the manuscript technically sound, and do the data support the conclusions?

Reviewer #1: Yes

Reviewer #2: Yes

3. Has the statistical analysis been performed appropriately and rigorously? 

Reviewer #1: Yes

Reviewer #2: Yes

4. Have the authors made all data underlying the findings in their manuscript fully available?

Reviewer #1: Yes

Reviewer #2: Yes

5. Is the manuscript presented in an intelligible fashion and written in standard English?

Reviewer #1: Yes

Reviewer #2: Yes

6. Review Comments to the Author

Reviewer #1: I support the publication of this manuscript. It has improved substantially. I would ask the authors to revise their title to into "in the Southeastern United States" because I believe a lot of key findings are very context specific to Florida. This will also be valuable for future readers who will ponder the unique COVID response in Florida as early as May 2020. It is important to note that college represents a diverse group of students and teachers who might share different opinions and attitudes from local authorities who removed mask policies at that time. It might be worth expanding on this as well.

- JMM RESPONSE

o Added “in the Southeastern United States” to the end of the title. 

o R1 suggests it may be worth adding more content about how/why students and teachers may have different opinions and attitudes from local authorities. 

I agree and add the following content:

• In particular, our sample consists of individuals linked to a university, whose mission is public education and research directed towards the public good. Furthermore, this university is in Leon County, a historically democratic population center, and it is likely that our sample universe embodied different values from conservative-leaning population centers in other parts of the state and country [4].

Reviewer #2: I think the article improved greatly in the revised version.

There are still some remaining issues which should be improved:

- There is a section in the introduction with the title “Previous research on COVID-19 and the U.S. Social and Political Response”. What is listed in this four short review paragraphs is only a very small snapshot of what has been researched on this topic, and it was not clear to my how the author selected the content. I think the title of the paragraph does not match the content. The authors should state explicitly at the beginning of this section that they selected some results that they think are relevant for the present research, or the title of the paragraph should me more specific.

- JMM RESPONSE

o Given the pace of publication on the COVID-19 pandemic, I really do believe it would be impossible to comprehensively cite all of the literature. 

For the Problem Orientation section, and the following subsection, I changed the title from “Previous Research on COVID-19…” to “Previous biosocial and structural research on COVID-19…” 

• This change in the subheading makes clearer why we selected certain publications to focus on for this section. As anthropologists, we tend to focus on biosocial approaches to health. 

- A formal definition of collectivism and individualism should be provided

- JMM RESPONSE

o Agreed and added the following text:

Collectivism describes a condition of society in which the needs of the group are prioritized over the individual; conversely, individualism prioritizes rights, concerns, needs, and desires of each individual [24–28]. Variations in individualism and collectivism are documented across the globe and these variations have influenced nation-state approaches to pandemic preparedness and response. 

- At the end of the introduction, the authors list some more research questions (is political affiliation related to COVID-19 safety behaviors; relationship between collectivism and trust in science). The authors should at least briefly introduce these questions with the idea behind it and the relevant references that addressed these questions already.

- JMM RESPONSE

o I’ve read through my introduction several times, and I just do not see what R2 is referring to here… there is no list of research questions in my intro. 

o There is a list of sub-inquiries (research questions) in my METHODS section. If these are the questions they meant, I have added a sentence explaining each research question and some citations. 

Hui and Yee report a positive association between age and collectivism, as well as between sex (females) and collectivism [46]. Vandello and Cohen report Florida in the 2nd quintile for collectivism due to the age of the population and a larger than average immigrant population [47]

Numerous studies have shown links between collectivism and safe COVID-19 behaviors [33,44,45,49,50], and we wanted to test this relationship through our sample and data. 

The political consequences of the pandemic have been described in detail [22,51,52], but we wanted to know if political affiliation influenced our study sample in their responses to the pandemic.

o Removed the trust in science inquiry, since we do not really focus on this in the article. 

- The first paragraph of the results section summarizes/discusses all relevant interpretations of the results. This is unusual, the results should be reported first.

- JMM RESPONSE

o I might argue this is a bit of a style issue. The first paragraph of our results section summarizes all of your results first, so that way the reader has the ability to grasp the significance of the results as they read through the rest of the section. Immediately after the first paragraph summary, we provide data reporting of all our results, with context and interpretation. I think our results are clearly presented, however, if R2 really wants me to create a separate section that simple just gives numerical results of the study, I’ve typed it out below, and it can be inserted into the RESULTS section as the very first paragraph. 

251 cases were analyzed for collectivism, yielding an average score of 91.4 (S.D. = 11.8), which is just slightly collectivist. We found that age, sex, and education are associated with collectivism in our sample. We also found that collectivism predicts for mask wearing, disrupted socializing, concern for infecting others, and trust in doctors regarding COVID-19 behavior guidelines. 

- A substantial part of the discussion contains results. The structure should be improved. The discussion should start with discussing the results of the main question, not with additional results

- JMM RESPONSE

o Our data analysis, models, and interpretation of results are all presented in the results section. What can’t be incorporated into that section are the written comments, which are qualitative, and difficult to synthesize. Consequently, we weave them into the discussion section. 

o I see R2’s point about improving structure and have revised this section, while also retaining some of the qualitative data. 

o Added – 

Our inquiry began with the question, how does an individual’s alignment along the collectivism/individualism scale impact their choices regarding the COVID-19 pandemic? Through a quantitative survey of FSU affiliates, we discovered age, sex, and education are associated with collectivism, and that individuals that are more collectivist also tend to wear masks, practice social distancing, have concerns for infecting others, and trust in doctors regarding COVID-19 behavior guidelines. Given our questionnaire also asked respondents questions about political affiliation, we wanted to unpack how and why political beliefs influenced our study samples, since numerous studies have shown these to be interlinked phenomena [52].

- There should be a participant section in the method section, where one can read number and demographic information (age, gender) of the participants for the different surveys. Were there really only 11 participants who conducted the semi-structured interview? Such limitations should also be mentioned in the discussion section.

- JMM RESPONSE

o I can see why R2 may want more data on the participants in the mixed methods survey, so I report on how many men and women we interviewed, but the interviews were not part of the analysis, only the research design to improve the survey tool. 

Added - (n=11 participants; 8 female, 3 male)

o TABLE 1, which is referenced in the RESULTS section, contains all of our demographic data for PHASE 2 Quantitative Survey Participants. 

o In the research design section, I made it clear that PHASE 1 Qualitative surveys were only used to refine the survey questionnaire, and not to interpret PHASE 2 data. Our semi-structured interviews were not really analytical components of the quantitative survey, but rather a tool we used to refine our survey instrument. 

o Added this text - 

Phase 1 data were not utilized in the analysis of phase 2 data, nor were respondent answers used to assist in the interpretation of phase 2 data. Rather, the semi-structured interviews were simply used as a tool to refine and improve the survey questionnaire, which is the basis for our analysis.

7. PLOS authors have the option to publish the peer review history of their article (what does this mean?). If published, this will include your full peer review and any attached files.

Do you want your identity to be public for this peer review? For information about this choice, including consent withdrawal, please see our Privacy Policy.

Reviewer #1: No

Reviewer #2: No

---

## [Editor Report · Decision Letter 2]

24 Nov 2022

Assessing the Role of Collectivism and Individualism on COVID-19 Beliefs and Behaviors in the Southeastern United States

PONE-D-22-14363R2

Dear Dr. Mehta,

We’re pleased to inform you that your manuscript has been judged scientifically suitable for publication and will be formally accepted for publication once it meets all outstanding technical requirements.

Kind regards,

Angelo Moretti, Ph.D.

Academic Editor

PLOS ONE
---

## [Editor Report · Acceptance letter]

8 Dec 2022

PONE-D-22-14363R2 

Assessing the Role of Collectivism and Individualism on COVID-19 Beliefs and Behaviors in the Southeastern United States 

Dear Dr. Mehta:

I'm pleased to inform you that your manuscript has been deemed suitable for publication in PLOS ONE. Congratulations! Your manuscript is now with our production department. 

Kind regards, 

on behalf of

Dr. Angelo Moretti 

Academic Editor

PLOS ONE